# A Multi-SCALE Community Network-Based SEIQR Model to Evaluate the Dynamic NPIs of COVID-19

**DOI:** 10.3390/healthcare11101467

**Published:** 2023-05-18

**Authors:** Cheng-Chieh Liu, Shengjie Zhao, Hao Deng

**Affiliations:** School of Software Engineering, Tongji University, No. 1239, Siping Road, Shanghai 200092, China

**Keywords:** SEIQR model, community network, COVID-19, non-pharmaceutical intervention, transmission dynamics

## Abstract

Regarding the problem of epidemic outbreak prevention and control, infectious disease dynamics models cannot support urban managers in reducing urban-scale healthcare costs through community-scale control measures, as they usually have difficulty meeting the requirements for simulation at different scales. In this paper, we propose combining contact networks at different spatial scales to study the COVID-19 outbreak in Shanghai from March to July 2022, calculate the initial $R_{t}$ through the number of cases at the beginning of the outbreak, and evaluate the effectiveness of dynamic non-pharmaceutical interventions (NPIs) adopted at different time periods in Shanghai using our proposed approach. In particular, our proposed contact network is a three-layer multi-scale network that is used to distinguish social interactions occurring in areas of different sizes, as well as to distinguish between intensive and non-intensive population contacts. This susceptible–exposure–infection–quarantine–recovery (SEIQR) epidemic model constructed based on a multi-scale network can more effectively assess the feasibility of small-scale control measures, such as assessing community quarantine measures and mobility restrictions at different moments and phases of an epidemic. Our experimental results show that this model can meet the simulation needs at different scales, and our further discussion and analysis show that the spread of the epidemic in Shanghai from March to July 2022 can be successfully controlled by implementing a strict long-term dynamic NPI strategy.

## 1. Introduction

In March 2022, an outbreak of COVID-19 caused by the Omicron variant of SARS-CoV-2 was reported in Shanghai, China. To contain the more contagious Omicron variant, Shanghai implemented a rigorous set of non-pharmaceutical interventions (NPIs) that include requirements to maintain social distancing, wear masks, track close contacts, isolate infected individuals, etc. [1,2]. A street grid was implemented from mid-March onwards, dividing the streets into high-risk and non-high-risk areas based on the number of infected individuals, and a comprehensive nucleic acid test screening was conducted. To more effectively control the spread of the epidemic, a city closure was initiated on 1 April 2022, and the epidemic reached an inflection point on 13 April, when the epidemic was eventually brought under control. Although the implementation of NPIs can successfully reduce the spread of COVID-19 [3], it also inevitably results in severe productivity losses and a substantial increase in social costs [4].

Many studies have been conducted using machine learning models (e.g., long short-term memory (LSTM) [5], neural networks [6]), or mathematical models (e.g., SIR [7], SEIR [8], and SEIQR [9]) to simulate COVID-19 transmission in various locations.

Machine learning models have been successfully applied in various domains. However, they are prone to overfitting in the presence of insufficient training data and valuable features. Machine learning models are also not suitable for heavy missing data scenarios [10], which are only eligible for short-term forecasting trends [11,12].

The SIR and SEIR mathematical models are the most common methods used in COVID-19 simulations [13]. The models divide the population into different compartments, and the movement of the population from one compartment to another is predicted by differential equations. The approach is flexible as the number of compartments can be varied. So far, there have been many studies that have improved the SEIR model, such as considering hospitalized individuals in the SEIHR model [14] and considering deaths in the SEIRD model [15], considering quarantine in the SEIQR model [9,16,17]. There have also been studies that incorporated various factors in the model, such as vulnerable populations [18], different severity symptoms [19], seasonal [20,21] and temperature [22] effects.

Mathematical models can simulate numerous variables that affect transmission to evaluate the severity of a pandemic and the effectiveness of interventions. However, such mathematical models are mostly deterministic and cannot respond to changes in different control strategies [23]. Therefore, some studies have evaluated the impact on COVID-19, the health care system, and society under various vaccination and NPI strategies by combining contact network modeling [24,25] for different age groups. However, these models do not reflect the variation in control strategies at different spatial scales. For example, when implementing quarantine measures, it is not possible to only quarantine people of a specific age group.

To address these issues, we propose a network model that allows us to easily assess the optimal population restriction strategy during the initial phase of an outbreak. This allows us to understand the impact of the strategy on the final size of the outbreak. The main contributions of this paper are as follows:1.We propose an SEIQR model that combines a multi-scale network, dividing the community network into three layers of different sizes, that can be used to distinguish social interactions occurring in administrative areas of different sizes and NPI strategies at different times.2.We divided the contacts formed in the community network into global transmission and local transmission, and used this method to distinguish between intensive and non-intensive crowd contacts.3.We simulated and analyzed the data from March to July 2022 in Shanghai and simulated each district in Shanghai separately. The results showed that the method can be used for both large- and small-scale simulations.

The rest of the paper is organized as follows: Section 2 provides an overview of the SEIQR model and community network; Section 3 analyzes the experimental results; Section 4 discusses the experimental results; finally, the paper is summarized in Section 5.

## 2. Methods

### 2.1. Data Collection

Three kinds of data were used in this research. First, the number of districts and population data were collected for Shanghai from the National Bureau of Statistics of China [26]. Second, we collected data on reported COVID-19 human cases from 1 March to 1 July, including cumulative cases, new daily cases, recovered cases, and deaths from the Shanghai Municipal Health Commission [27]. Third, we tracked the policies implemented, such as contact tracing and quarantine, quarantine of areas concerned, and the implementation time schedule and scope of PCR testing [28]. These data are publicly available.

### 2.2. SEIQR Model

The typical SEIQR model classifies each individual into five categories: susceptible (*S*), exposed (*E*), infected (*I*), quarantined (*Q*), and recovered (*R*). It was considered that NPIs in Shanghai quarantine infect individuals and close contacts of infected individuals. A close contact is defined as a person who has not taken effective protection and has close contact (within 1 m) with confirmed and suspected patients 2 days before symptom onset in suspected and confirmed patients or 2 days before samples of asymptomatic infection are taken [29]. We used an improved version of the method with reference to [30]. The new model divides quarantined patients into two states: “exposed quarantined” ($Q_{E}$) and “infected quarantined” ($Q_{I}$). These two states represent quarantine of asymptomatic close contacts and quarantine of infected individuals, respectively. In a closed system that does not consider entries or exits, the sum of these intervals $N=S\left(t\right)+E\left(t\right)+I\left(t\right)+Q_{E}\left(t\right)+Q_{I}\left(t\right)+R\left(t\right)$ remains constant in time. These kinetic models consist of the following equations:(1)$$\begin{array}{cc} \frac{dS}{dt} & =\frac{-\beta SI}{N}-\frac{\beta qSQ_{I}}{N}\\ \frac{dE}{dt} & =\frac{\beta SI}{N}+\frac{\beta qSQ_{I}}{N}-\sigma E-\theta_{E}E\\ \frac{dI}{dt} & =\sigma E-\gamma I-\theta_{I}I\\ \frac{dQ_{E}}{dt} & =\theta_{E}E-\sigma_{Q}Q_{E}\\ \frac{dQ_{I}}{dt} & =\theta_{I}I+\sigma_{Q}Q_{E}-\gamma_{Q}Q_{I}\\ \frac{dR}{dt} & =\gamma I+\gamma_{Q}Q_{I} \end{array}$$

The fraction $\frac{SI}{N}$ represents the probability of arbitrary contact between susceptible and infected individuals, and the fraction $\frac{SQ_{I}}{N}$ corresponds to the probability of contact between susceptible and isolated infected individuals. The main parameters of the model can be seen in Table 1 below. The parameter *q* was used to indicate the probability that the isolated individual interacts with others. We set this parameter using the difference in the number of contacts counted by [31]. In [31], contact survey data in three different phases of the pandemic (before the pandemic, during the lockdown, and post-lockdown) in four locations in China (Wuhan, Shanghai, Shenzhen, and Changsha) were collected. Survey data show that the average daily contacts in Shanghai before the pandemic was 18.8 (95% CI 17.5–20.1), and would have been reduced to 2.3 (95% CI 2–2.8) during the lockdown. We used this data to set *q* = 0.1. The transition rate of people’s exposure takes the form $\frac{dS}{dt}=\frac{-\beta SI}{N}-\frac{\beta qSQ_{I}}{N}$. The probability of moving from the exposure stage to the infection stage is represented by σ. The parameter θ denotes the probability that an individual is isolated. The final probability of moving from the infection stage to the recovery stage is the γ parameter. We will adapt this model in several different ways to include recent epidemiological information.

The parameter β (i.e., infection rate) of an individual is equal to the expected number of cases generated by that individual in a fully susceptible population (i.e., the time-dependent reproduction number $R_{t}$ of that individual divided by the length of its infection period α).
(2)$$\begin{array}{c} \beta =\frac{R_{t}}{\alpha } \end{array}$$

The reproduction number (*R*) [32] is one of the most critical parameters determining disease dynamics within infectious disease models, providing a summary measure of the transmission potential. The actual average number of secondary cases per infected person at time t is called the time-dependent reproduction number ($R_{t}$). $R_{t}$ shows the temporal changes that occur with the implementation of control measures, where a value greater than 1 indicates that the epidemic will be self-sustaining and a value less than 1 indicates that the number of new cases will decrease over time and that the epidemic will eventually stop. We used the method proposed by [33] to calculate the $R_{t}$ values in our model from the data at the beginning of the outbreak.

### 2.3. Community Network

Considering that non-pharmaceutical intervention policies at different administrative area scales are implemented at the beginning of an outbreak, we constructed a multi-scale community network with different administrative area scales based on the FARZ algorithm [34] as shown in Figure 1 and described below. The FARZ algorithm proposes a new method to evaluate the accuracy of community detection algorithms using modular networks. Community detection algorithms are used to partition a network into subsets with intrinsic similarities, such as dividing users in a social network into different communities. However, the accuracy of these algorithms is difficult to assess precisely because the true community structure is often unknown. The FARZ algorithm proposes a method based on modular networks to address this issue, which involves partitioning the network into different modules and evaluating accuracy by comparing the similarity between these modules and pre-defined benchmark modules. The FARZ algorithm expands the network one node at a time using a probability of community assignment proportional to the current community size. The resulting network has a heavy-tailed distribution with similar characteristics to the real network. FARZ has three input parameters, FARZ (n, m, k), which determine the population, average number of contacts, and number of communities, respectively. It also has four control parameters that control the strength of the community structure, clustering coefficient, degree correlation, and distribution of the community sizes, respectively, with values taken from [35].

Specifically, we divided the contact network into three layers and used three sizes of areas to perform the division of close contacts: the number of administrative districts (large range), number of townships (medium range), and number of residential areas (small range) in Shanghai, respectively. In the papers [36,37], the relationship between walking behavior and districts or environmental factors is studied. The different contact weights in the three network layers were set by referring to the duration and importance percentages of the three types of walking proposed by [38]. The paper uses survey data from Shanghai to assess the relationship between environmental characteristics and three domains of walking behavior (commuting, utilitarian walking, and recreational walking). Commuting walking indicates commuting trips from home to work and includes the duration and details of the mode of transportation (e.g., car, walking, biking, motorcycle, metro, or bus). Utilitarian walking includes walking for non-commuting travel purposes in order to satisfy daily needs (e.g., shopping, visiting friends, and running errands) [36]. Recreational walking is defined as walking, strolling, jogging, and other walking behaviors that focus on walking itself. The average weekly duration of the three walking behaviors was 124.99 min (standard deviation (SD) ± 130.03), 51.02 min (SD ± 204.45), and 36.14 min (SD ± 96.76), respectively. The relative importance of built environment attributes (e.g., distance) and individual-level attributes (e.g., age) is also summarized in the paper [38], and only the built environment importance was considered in our study. The importance of the built environment for the three walks was 75.6%, 86.2%, and 81.6%, respectively.

As shown in Figure 2, the three layers have their own contact weights in the network, which are calculated as (average weekly movement time/total average weekly movement time) × the importance of the built environment. $B_{A}$ corresponds to commuting walking, calculated as (124.99/212.15) × 0.756 = 0.445. $B_{T}$ corresponds to utilitarian walking, calculated as (51.02/212.15) × 0.86 = 0.207. $B_{R}$ corresponds to recreational walking, calculated as (36.14/212.15) × 0.816 = 0.139. The three range settings through the three-layer network can reflect different variations in NPIs and can also show different trends in infection behavior under the restriction of different movement patterns. For example, when Beijing took NPIs to prevent the deterioration of the disease in the early stage of COVID-19 [39], people were restricted to their respective administrative districts; thus, the infection behavior was easier to manage and control in a large range compared with possible negligence and omission in a small range. Therefore, it is clear that NPI strategies for different tiers are tailored to the characteristics of the tier scale, thus allowing for a more effective handling of COVID-19.

The dynamics of controlled transmission events leading to the exposure of susceptible individuals are the basis of epidemiological models; thus, here, we explain the transmission dynamics of the model in detail. For the model considered in this study, the propensity of a given individual to become infected was calculated using the following equation, where disease transmission to close contacts or to incidental contacts may occur, as determined by the structure of the exposure network. Local transmission ($T_{L}$) is a person who is in frequent repeated continuous or close contact: a family member, friend, or other close relationship. In contrast, global transmission ($T_{G}$) are infrequent casual or brief contacts.

The model’s differential equations were applied to calculate the propensity of the possible state transitions at all nodes at each time step. These propensities were then used to compute the probabilities of all possible state transitions normalized across the entire population. A random node and the corresponding transition were selected in each time step based on these probabilities. Each node *i* has an indicator function $\delta_{X}\left(i\right)$ that represents the state of node *i*. If node *i* is in *X*, its value is 1, otherwise the value is 0.
(3)$$\begin{array}{c} \delta_{X}\left(i\right)=\left\{\begin{array}{cc} 1 & \mathrm{if}\;i\in X\\ 0 & \mathrm{if}\;i\notin X \end{array}\right. \end{array}$$

In general, the propensity to infect susceptible individual *i*, $P^{\left(i\right)}\left(S\to E\right)$, is proportional to the association between the prevalence and incidence of infected individuals in the exposed population, and the propensity to transition between states for a given node is given by the following equation:(4)$$\begin{array}{cc} P^{\left(i\right)}\left(S\to E\right) & =\; [\left(1-B_{A}-B_{T}-B_{R}\right)T_{G}+\left(B_{A}+B_{T}+B_{R}\right)T_{L}]\delta_{S}\left(i\right)\\ P^{\left(i\right)}\left(E\to I\right) & =\;\sigma \delta_{E}\left(i\right)\\ P^{\left(i\right)}\left(I\to R\right) & =\;\gamma \delta_{I}\left(i\right)\\ P^{\left(i\right)}\left(E\to Q_{E}\right) & =\;\theta_{E}\delta_{E}\left(i\right)\\ P^{\left(i\right)}\left(I\to Q_{I}\right) & =\;\theta_{I}\delta_{I}\left(i\right)\\ P^{\left(i\right)}\left(Q_{E}\to Q_{I}\right) & =\;\sigma_{Q}\delta_{Q_{E}}\left(i\right)\\ P^{\left(i\right)}\left(Q_{I}\to R\right) & =\;\gamma_{Q}\delta_{Q_{I}}\left(i\right) \end{array}$$

#### 2.3.1. Global Transmission

A portion of a given individual’s interactions is with casual contacts that are assumed to be individuals randomly selected from the total population, independent of the contact network. For these global interactions, each node in the population is equally likely to be in contact with every other node, and the population can be considered to be well-mixed. Thus, the global transmission ($T_{G}$) is calculated in the same way as the mass action interval model, assuming that the population is well-mixed.
(5)$$\begin{array}{c} T_{G}=\frac{\beta I+\beta qQ_{I}}{N} \end{array}$$

#### 2.3.2. Local Transmission

Some of the interactions of a particular individual are with their “close contacts”. The close contacts of an individual are defined as the nodes adjacent to a given node in the contact network, and $C_{G}\left(i\right)$ denotes the set of close contacts of individual *i*: the nodes adjacent to node *i* in the contact network graph G. $|C_{G}\left(i\right)|$ denotes the size of this set: the number of close contacts owned by *i*.

With respect to local transmission, transmissibility is considered on a pairwise basis; that is, an infection rate β is assigned to each directed edge of the contact network representing the propagation from infected node *j* to susceptible node *i*, which depends on the infection rate of a single infected individual *j*. The propensity for a given susceptible individual to become exposed due to local transmission is calculated as the product of that individual’s susceptibility and the sum of the transmissibility of their infectious close contacts divided by the size of their local network.
(6)$$\begin{array}{c} T_{L}=\frac{\beta [\sum_{j\in C_{G}\left(i\right)}\delta_{I}\left(j\right)]+\beta q [\sum_{k\in C_{G}\left(i\right)}\delta_{Q_{i}}\left(k\right)]}{\left|C_{G}\left(i\right)\right|} \end{array}$$

### 2.4. Global Sensitivity Analysis

Global sensitivity analysis (GSA) apportions the influence that model parameter uncertainties have on the uncertainty of model output [40]. In this analysis, GSA is performed using the Sobol’ method [41], which is a variance-based method that provides a measure of the sensitivity of the output to each input parameter. The Sobol’ indices used in this report are first-order sensitivity indices (S1) and total sensitivity indices (ST). The first-order sensitivity indices measure the effect of the input parameter on the output when the other parameters are fixed, while the total sensitivity indices measure the effect of the input parameter on the output when the other parameters are varied along with it.

In order to perform the sensitivity analysis, we use the Saltelli sampling method provided by the SALib package, which generates a set of input parameter values. The output of the model for each set of input parameter values is then used to compute the Sobol’ indices, which measure the sensitivity of the output to each of the input parameters. The Sobol’ indices are used to rank the input parameters by their importance in determining the output.

In more detail, the Saltelli sample is generated from the continuous input variables in the problem dictionary using a uniform distribution. In this case, the problem dictionary specifies nine input variables with bounds between 0 and 1. The sample size is set to 1024 and the input variables are sampled using the Saltelli method, which generates N × (2 × D + 2) samples, where N is the number of unique values for each input variable and D is the number of input variables. For this analysis, N is set to 8 (a power of 2) resulting in 1536 samples drawn from uniform distributions for each variable. The Saltelli sample is then used to generate the model output, which is the number of infected individuals for each parameter combination. The Sobol’ indices are then computed using the model output and the Saltelli sample to determine the relative importance of each input variable.

## 3. Results

### 3.1. Epidemic Curves for Shanghai

We conducted simulations for Shanghai, and the model parameters can be seen in Table 2 below. Figure 3 shows the long-term dynamics of the model, taking into account the closure measures implemented in Shanghai. Shanghai began implementing strict closure measures on 1 April and gradually eased the quarantine policy from 16 May. The model predicts that the peak number of infections occurred in mid-April. The trend between the curve simulated by the model (red line) and the real data (yellow line) is close as can be seen in the figure.

We compared our model (red line) and the model using the population age distribution (blue line) with the real data (yellow line), and the difference between the two models and the real data can be seen in Figure 4. The population age distribution model [35] was used to construct community networks with different age distributions and household sizes. We collected data on the age distribution [26] and the number of contacts in each age group [31] in Shanghai and simulated the infection results from March to July in Shanghai. We also calculated the median absolute percentage error (MAPE) between the two models and the real data. In the case of the 4-month-long simulation, it is reduced by 13.08% compared to the population age distribution model. The MAPE of 24.96% for the multi-scale network model in the strict closure measure time period (4/1–5/16) is also lower than that of 40.12% for the population age distribution model, showing a reduction of 15.16%.

### 3.2. Epidemic Curves for 16 Administrative Districts in Shanghai

To view the model simulations at a smaller scale and compare them with the simulation results of the entire city of Shanghai, we applied the model framework to 16 districts in Shanghai for individual simulations, and the model parameters can be seen in Table 3 below. For the district simulation, we reduced the number of layers in the community network, kept only the township district layer and residential areas layer, and set the initial parameters for different districts.

In Figure 5, we can see that the simulated infection trends are partially consistent for each district, especially in areas with more confirmed cases, such as Minhang and Pudong, but there are also areas where the simulated results are not consistent, which is due to two reasons: there are two peaks in the actual number of confirmed cases during the quarantine period, such as Hongkou, or the actual number of confirmed cases is too small and the transmission is successfully suppressed at an early stage, such as Jinshan.

As shown in Figure 6, we summed up the number of infections simulated in the 16 districts (black line) and compared it with the simulation results in Shanghai (red line). We found that the MAPE between the two was 32.02%. The obvious difference between the two results lies in the peak of the wave, which we believe is mainly due to the fact that the 16 districts were simulated separately and that there was no interaction between the districts before the implementation of the control measures, i.e., before 1 April. Thus, the peak size of the outbreak could not reach as high as the Shanghai simulation results.

### 3.3. Global Sensitivity Analysis of the SEIQR Model


In this report, we present the results of a global sensitivity analysis (GSA) of the SEIQR model, using the Sobol’ method, and the results can be seen in Table 4 below. The analysis aimed to determine the impact of each of the model’s input parameters on the uncertainty of the model output. The model takes into account the spread of a hypothetical disease in a population, and the output is the number of infected individuals. The Sobol’ indices used in this report are first-order sensitivity indices (S1) and total sensitivity indices (ST). The first-order sensitivity indices measure the effect of the input parameter on the output when the other parameters are fixed, while the total sensitivity indices measure the effect of the input parameter on the output when the other parameters are varied along with it.

The results of the analysis show that the most influential parameter on the model’s output is γ, the recovery rate from the infectious state, with a total sensitivity index of 1.86. This means that when all input parameters are varied simultaneously, γ has the greatest impact on the model output. The second most influential parameter is $R_{t}$, the basic reproduction number, with a total sensitivity index of 0.11. The third most influential parameter is σ, the rate of latent individuals becoming infectious, with a total sensitivity index of 0.22. The first-order sensitivity indices (S1) show the effect of each input parameter on the output when the other parameters are fixed. The results indicate that the parameter with the greatest first-order sensitivity index is γ, followed by $R_{t}$ and $\theta_{I}$, the rate of recovery from the infectious state. This means that these parameters have a relatively large influence on the model output, even when the other parameters are held constant.

In summary, the results of the sensitivity analysis suggest that the most important parameters to consider when trying to reduce the uncertainty in the SEIQR model output are γ, $R_{t}$, and σ. These parameters should be the focus of future research and efforts to improve the model’s accuracy.

## 4. Discussion

Since the strength of mathematical models, whether simulating or predicting future dynamics, depends on the accuracy of the case reports, it is important to analyze the uncertainty and to assess the degree of variability in the parameter estimates. The time span of the time series used can significantly affect the values of the parameters. This is particularly true in the early stages of an epidemic, when the relationship between infected, deceased, and recovered individuals is not entirely clear. For example, if the number of confirmed cases is low, it is difficult to determine whether quarantine measures are being strictly enforced. Moreover, this under-representation suggests that the number of infections is much larger than the number of confirmed cases. Among the most affected parameters are those related to the infection and recovery rates and the $R_{t}$ value.

It can be concluded that the smaller the area from which quarantine between populations is implemented, the more effectively the spread of the epidemic is limited. However, such a quarantine from a small area will result in a high medical cost, in addition to inconvenience to people’s lives, so it is important to strike a balance between the extent of the quarantine and the social cost.

Any such modeling study has many limitations [43]. An extended description of the limitations specific to this study is provided. Specifically, (1) these models are only approximations of real-world scenarios, and we simplified many aspects of the epidemiological process of disease transmission in order to make these models computationally feasible. For example, we did not consider any entries or exits for closed systems and did not consider effects such as vaccination differences [44] or seasonal factors [22]; (2) these models are strongly driven by infection rate data, whose recording can be problematic; (3) these models are also influenced by a variety of other data types, each with different availabilities, and we may never be able to fully calibrate detection and measurement bias issues; and (4) the understanding of this dynamic epidemic is growing daily, so this modeling framework should not be expected to be definitive, or the data driving these models should not be expected to be fixed. Finally, we emphasize that we modeled a range of outcomes that we believe are likely to occur under the scenario based on the data observed to date and our model assumptions. We recommend these simulations to be viewed as useful guides.

Additionally, we made the simplifying assumption that districts are isolated from each other and that the only way for infection to spread between districts is through global transmission or contacts in other network layers. However, in reality, there is often significant inter-regional mobility that can impact the spread of infectious diseases [45]. Future studies could explore how incorporating mobility between regions may affect our model’s predictions.

## 5. Conclusions

In this study, we implemented a modified SEIQR model that incorporates a multi-scale network that considers social interactions occurring at different spatial scales while distinguishing between intensive and non-intensive crowd contacts in order to assess the impact of NPIs on the epidemic at different time periods. We used infection data from March 2022 to July 2022 in Shanghai for the simulation and analysis, and the experimental results show that the model is able to successfully simulate infection outcomes over a long period of time using pre-infection data and can be applied to simulations of different scales and NPIs. When facing the emergence of new variants in the future, interventions in different countries and regions may have different degrees of impact. Our model can provide useful recommendations for preventing and controlling COVID-19 outbreaks.

## Figures and Tables

**Figure 1 healthcare-11-01467-f001:**
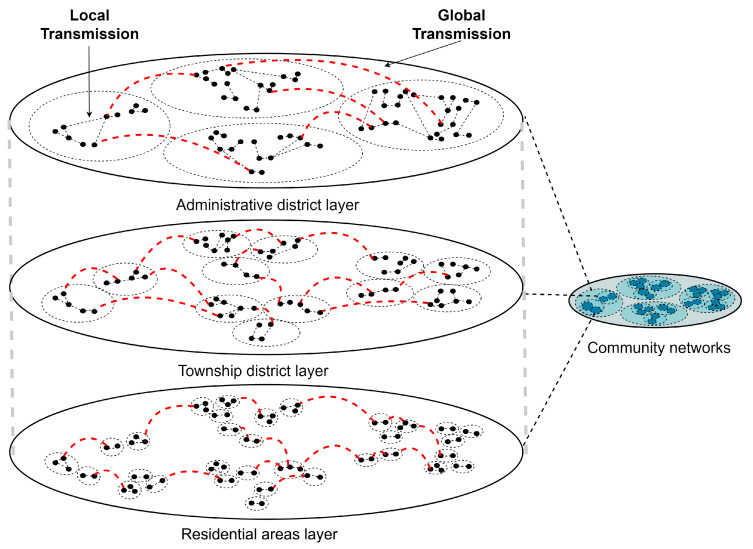
Community network layered framework diagram.

**Figure 2 healthcare-11-01467-f002:**
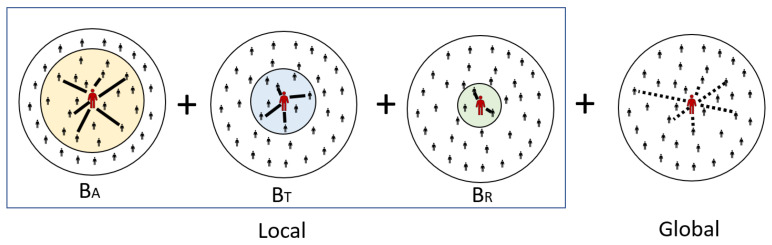
Local and global transmission diagram.

**Figure 3 healthcare-11-01467-f003:**
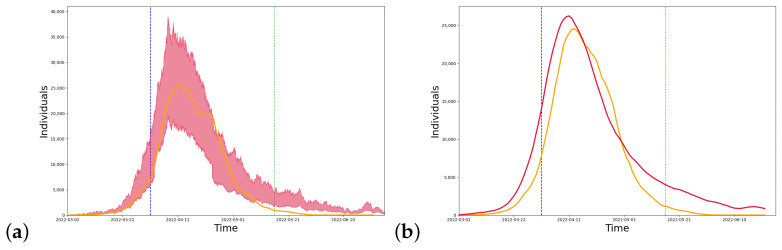
(**a**) Simulation results. (**b**) After numerical smoothing. The simulation curve for Shanghai from 2 March, where the red line is the model simulation result, the yellow line is the real number of confirmed cases, the blue dotted line is the start of the closure on 1 April, and the green dotted line is the time when the closure measures were slowed down on 16 May.

**Figure 4 healthcare-11-01467-f004:**
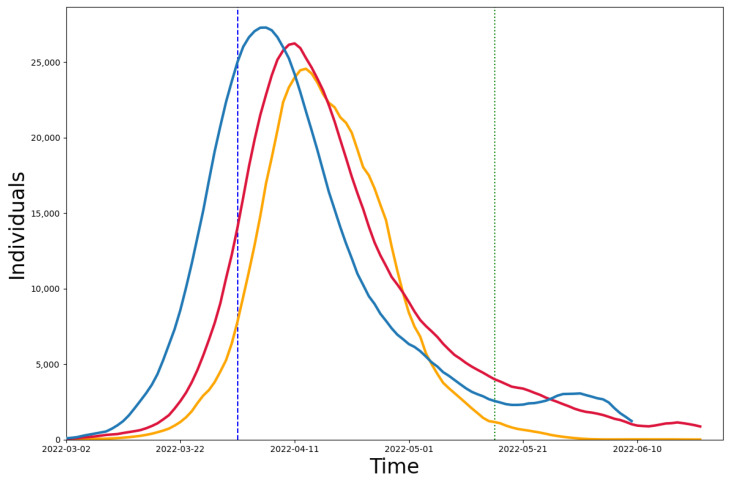
Comparison between models. The red line is the result of the multi-scale network model simulation, the yellow line is the real number of confirmed diagnoses, and the blue line is the infection curve simulated using the Shanghai population age distribution model.

**Figure 5 healthcare-11-01467-f005:**
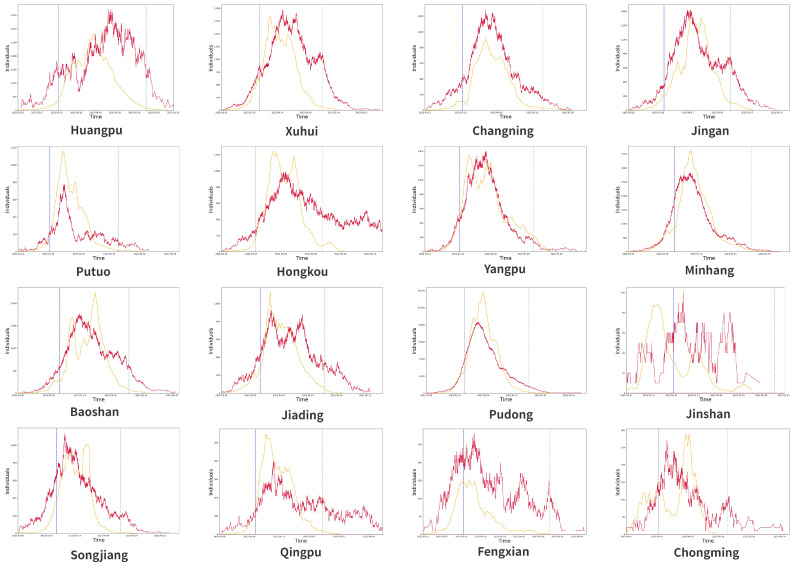
Simulation of the 16 administrative districts of Shanghai.

**Figure 6 healthcare-11-01467-f006:**
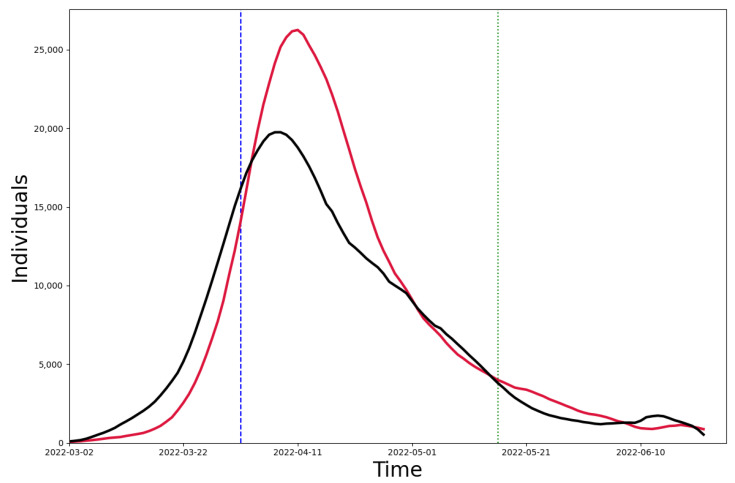
The black line is the total number of infections in the 16 administrative regions, and the red line is the simulation result for Shanghai.

**Table 1 healthcare-11-01467-t001:** Main model parameters description.

Parameter	Description
β	Infection rate
γ	Recovery rate
σ	Quarantine rate
θ	Morbidity rate
*q*	Quarantined infection weight
*N*	Total number of nodes
$R_{t}$	Reproduction number
α	Infection period

**Table 2 healthcare-11-01467-t002:** Model parameters for March to July in Shanghai.

Parameter	Values
$B_{A}$	0.445
$B_{T}$	0.207
$B_{R}$	0.139
$R_{t}$	4.6 (95% CI 3.97–5.36)
α	7.7 (95% CI 7.26–8.15) [42]
γ	0.38
σ	0.967
θ	0.02
*q*	0.1
Population ($10^{3}$)	24,900
Number of administrative districts	16
Number of township districts	215
Number of residential areas	6228
Initial number of infections	2
Start time	3/2
Quarantine start time	4/1
Quarantine relief time	5/16

**Table 3 healthcare-11-01467-t003:** Description of the parameters of the 16 administrative regions.

Community	$R_{t}$	Initial Number of Infections	Start Time	Population $\left(10^{3}\right)$	Number of Township Districts	Number of Residential Areas
Huangpu	2.55	5	3/11	662	10	169
Xuhui	3.0	3	3/5	1113	13	307
Changning	3.27	1	3/11	693	10	185
Jingan	3.27	2	3/8	975	14	266
Putuo	4.03	1	3/11	1239	10	275
Hongkou	2.6	4	3/9	757	8	194
Yangpu	4.2	2	3/11	1242	12	288
Minhang	3.7	6	3/5	2653	13	586
Baoshan	3.32	2	3/5	2235	12	509
Jiading	2.44	4	3/5	1834	10	378
Pudong	4.87	2	3/5	5681	36	1398
Jinshan	2.14	4	3/11	822	10	238
Songjiang	2.8	7	3/5	1907	17	367
Qingpu	1.98	3	3/8	1271	11	351
Fengxian	3.27	1	3/11	1140	11	340
Chongming	2.85	3	3/11	637	18	377

**Table 4 healthcare-11-01467-t004:** Global sensitivity analysis.

Parameter	S1	ST
$R_{t}$	0.0419	0.1912
Infectious period (days)	0.0384	0.1077
σ	0.0445	0.1952
γ	0.1420	1.1236
$\sigma_{Q}$	0.0006	0.0002
$\gamma_{Q}$	0.0053	0.0356
$\theta_{E}$	0.0702	0.2218
$\theta_{I}$	0.1698	1.8588
*q*	0.0137	0.0209

## Data Availability

Data will be provided by the authors on request.

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
