# Peer review of "A Multi-SCALE Community Network-Based SEIQR Model to Evaluate the Dynamic NPIs of COVID-19"

_healthcare, 2023, doi:10.3390/healthcare11101467_

Round 1

Reviewer 1 Report

This article proposes a model to describe covid19 infection, as we know, since the outbreak of covid19, researchers have proposed many models to describe its infection process. I think the authors should discuss the advantages and disadvantages of their model compared to other models.

The author used a 3-layer network to transform the interpersonal interaction into local and global transmission, but independently simulated each of the 16 regions. I think the author did not fully reflect the role of the network, and its role in parameter estimation is completely used survey instead.

In addition, although the FARZ model mentioned in the article is described in the reference, it is beneficial to give a brief introduction in the article.

Reviewer 2 Report

The paper misses out on quoting several similar articles that are using the same approach and data analysis. Some may have been quoted but a major revision is required to establish the value of the research:

1: Wang Y, Wang P, Zhang S, Pan H. Uncertainty Modeling of a Modified SEIR

Epidemic Model for COVID-19. Biology (Basel). 2022 Aug 2;11(8):1157. doi:

10.3390/biology11081157. PMID: 36009784; PMCID: PMC9404969.

2: Bai T, Wang D, Dai W. A modified SEIR model with a jump in the transmission

parameter applied to COVID-19 data on Wuhan. Stat (Int Stat Inst). 2022

Dec;11(1):e511. doi: 10.1002/sta4.511. Epub 2022 Dec 23. PMID: 36713680; PMCID:

PMC9874617.

3: Youssef HM, Alghamdi NA, Ezzat MA, El-Bary AA, Shawky AM. A modified SEIR

model applied to the data of COVID-19 spread in Saudi Arabia. AIP Adv. 2020 Dec

4;10(12):125210. doi: 10.1063/5.0029698. PMID: 33304643; PMCID: PMC7722269.

4: Li W, Gong J, Zhou J, Zhang L, Wang D, Li J, Shi C, Fan H. An evaluation of

COVID-19 transmission control in Wenzhou using a modified SEIR model. Epidemiol

Infect. 2021 Jan 8;149:e2. doi: 10.1017/S0950268820003064. PMID: 33413715;

PMCID: PMC7804084.

5: Angulo W, Ramírez JM, De Cecchis D, Primera J, Pacheco H, Rodríguez-Román E.

A modified SEIR model to predict the behavior of the early stage in coronavirus

and coronavirus-like outbreaks. Sci Rep. 2021 Aug 11;11(1):16331. doi:

10.1038/s41598-021-95785-y. PMID: 34381100; PMCID: PMC8357795.

6: López L, Rodó X. A modified <i>SEIR</i> model to predict the COVID-19

outbreak in Spain and Italy: Simulating control scenarios and multi-scale

epidemics. Results Phys. 2021 Feb;21:103746. doi: 10.1016/j.rinp.2020.103746.

Epub 2020 Dec 25. PMID: 33391984; PMCID: PMC7759445.

7: Das A, Dhar A, Goyal S, Kundu A, Pandey S. COVID-19: Analytic results for a

modified SEIR model and comparison of different intervention strategies. Chaos

Solitons Fractals. 2021 Mar;144:110595. doi: 10.1016/j.chaos.2020.110595. Epub

2021 Jan 5. PMID: 33424141; PMCID: PMC7785284.

8: Li Y, Hou S, Zhang Y, Liu J, Fan H, Cao C. Effect of Travel Restrictions of

Wuhan City Against COVID-19: A Modified SEIR Model Analysis. Disaster Med Public

Health Prep. 2022 Aug;16(4):1431-1437. doi: 10.1017/dmp.2021.5. Epub 2021 Jan 8.

PMID: 33413723; PMCID: PMC8027550.

9: Ahmad N. COVID-19 Modeling in Saudi Arabia Using the Modified Susceptible-

Exposed-Infectious-Recovered (SEIR) Model. Cureus. 2020 Sep 14;12(9):e10452.

doi: 10.7759/cureus.10452. PMID: 33072460; PMCID: PMC7557707.

10: Cao S, Feng P, Shi P. [Study on the epidemic development of COVID-19 in

Hubei province by a modified SEIR model]. Zhejiang Da Xue Xue Bao Yi Xue Ban.

2020 May 25;49(2):178-184. Chinese. doi: 10.3785/j.issn.1008-9292.2020.02.05.

PMID: 32391661; PMCID: PMC8800716.

11: Mwalili S, Kimathi M, Ojiambo V, Gathungu D, Mbogo R. SEIR model for

COVID-19 dynamics incorporating the environment and social distancing. BMC Res

Notes. 2020 Jul 23;13(1):352. doi: 10.1186/s13104-020-05192-1. PMID: 32703315;

PMCID: PMC7376536.

12: Wali M, Arshad S, Huang J. Stability Analysis of an Extended SEIR COVID-19

Fractional Model with Vaccination Efficiency. Comput Math Methods Med. 2022 Sep

20;2022:3754051. doi: 10.1155/2022/3754051. PMID: 36176740; PMCID: PMC9514930.

13: Chen Y, Liu F, Yu Q, Li T. Review of fractional epidemic models. Appl Math

Model. 2021 Sep;97:281-307. doi: 10.1016/j.apm.2021.03.044. Epub 2021 Apr 20.

PMID: 33897091; PMCID: PMC8056944.

14: Maged A, Ahmed A, Haridy S, Baker AW, Xie M. SEIR Model to address the

impact of face masks amid COVID-19 pandemic. Risk Anal. 2023 Jan;43(1):129-143.

doi: 10.1111/risa.13958. Epub 2022 Jun 15. PMID: 35704273; PMCID: PMC9349537.

15: Ding X, Huang S, Leung A, Rabbany R. Incorporating dynamic flight network in

SEIR to model mobility between populations. Appl Netw Sci. 2021;6(1):42. doi:

10.1007/s41109-021-00378-3. Epub 2021 Jun 10. PMID: 34150986; PMCID: PMC8205202.

16: Das T, Srivastava PK, Kumar A. Nonlinear dynamical behavior of an SEIR

mathematical model: Effect of information and saturated treatment. Chaos. 2021

Apr;31(4):043104. doi: 10.1063/5.0039048. PMID: 34251223.

17: Jung S, Kim JH, Hwang SS, Choi J, Lee W. Modified susceptible-exposed-

infectious-recovered model for assessing the effectiveness of non-pharmaceutical

interventions during the COVID-19 pandemic in Seoul. J Theor Biol. 2023 Jan

21;557:111329. doi: 10.1016/j.jtbi.2022.111329. Epub 2022 Oct 26. PMID:

36309117; PMCID: PMC9598254.

18: Weinstein SJ, Holland MS, Rogers KE, Barlow NS. Analytic solution of the

SEIR epidemic model via asymptotic approximant. Physica D. 2020 Oct;411:132633.

doi: 10.1016/j.physd.2020.132633. Epub 2020 Jun 25. PMID: 32834248; PMCID:

PMC7316071.

19: Ruan S. Modeling the transmission dynamics and control of rabies in China.

Math Biosci. 2017 Apr;286:65-93. doi: 10.1016/j.mbs.2017.02.005. Epub 2017 Feb

8. PMID: 28188732; PMCID: PMC7094565.

20: Bongolan VP, Minoza JMA, de Castro R, Sevilleja JE. Age-Stratified Infection

Probabilities Combined With a Quarantine-Modified Model for COVID-19 Needs

Assessments: Model Development Study. J Med Internet Res. 2021 May

31;23(5):e19544. doi: 10.2196/19544. PMID: 33900929; PMCID: PMC8168636.

21: Yang Z, Zeng Z, Wang K, Wong SS, Liang W, Zanin M, Liu P, Cao X, Gao Z, Mai

Z, Liang J, Liu X, Li S, Li Y, Ye F, Guan W, Yang Y, Li F, Luo S, Xie Y, Liu B,

Wang Z, Zhang S, Wang Y, Zhong N, He J. Modified SEIR and AI prediction of the

epidemics trend of COVID-19 in China under public health interventions. J Thorac

Dis. 2020 Mar;12(3):165-174. doi: 10.21037/jtd.2020.02.64. PMID: 32274081;

PMCID: PMC7139011.

22: Benthall S, Hatna E, Epstein JM, Strandburg KJ. Privacy and contact tracing

efficacy. J R Soc Interface. 2022 Sep;19(194):20220369. doi:

10.1098/rsif.2022.0369. Epub 2022 Sep 21. PMID: 36128709; PMCID: PMC9490344.

23: Morillas F, Valero J. Random resampling numerical simulations applied to a

SEIR compartmental model. Eur Phys J Plus. 2021;136(10):1067. doi:

10.1140/epjp/s13360-021-02003-9. Epub 2021 Oct 25. PMID: 34722097; PMCID:

PMC8542918.

24: Levin MW, Shang M, Stern R. Effects of short-term travel on COVID-19 spread:

A novel SEIR model and case study in Minnesota. PLoS One. 2021 Jan

22;16(1):e0245919. doi: 10.1371/journal.pone.0245919. PMID: 33481956; PMCID:

PMC7822539.

25: Xu C, Yu Y, Chen Y, Lu Z. Forecast analysis of the epidemics trend of

COVID-19 in the USA by a generalized fractional-order SEIR model. Nonlinear Dyn.

2020;101(3):1621-1634. doi: 10.1007/s11071-020-05946-3. Epub 2020 Sep 14. PMID:

32952299; PMCID: PMC7487266.

26: Eksin C, Ndeffo-Mbah M, Weitz JS. Reacting to outbreaks at neighboring

localities. J Theor Biol. 2021 Jul 7;520:110632. doi:

10.1016/j.jtbi.2021.110632. Epub 2021 Feb 25. PMID: 33639138; PMCID: PMC7904447.

27: Arfan M, Lashin MMA, Sunthrayuth P, Shah K, Ullah A, Iskakova K, Gorji MR,

Abdeljawad T. On nonlinear dynamics of COVID-19 disease model corresponding to

nonsingular fractional order derivative. Med Biol Eng Comput. 2022

Nov;60(11):3169-3185. doi: 10.1007/s11517-022-02661-6. Epub 2022 Sep 15. PMID:

36107356; PMCID: PMC9476458.

28: Aziz MHN, Safaruddin ADA, Hamzah NA, Supadi SS, Yuhao Z, Aziz MA. Modelling

the Effect of Vaccination Program and Inter-state Travel in the Spread of

COVID-19 in Malaysia. Acta Biotheor. 2022 Nov 17;71(1):2. doi:

10.1007/s10441-022-09453-3. PMID: 36394646; PMCID: PMC9670086.

29: Durai CAD, Begum A, Jebaseeli J, Sabahath A. COVID-19 pandemic, predictions

and control in Saudi Arabia using SIR-F and age-structured SEIR model. J

Supercomput. 2022;78(5):7341-7353. doi: 10.1007/s11227-021-04149-w. Epub 2021

Nov 10. PMID: 34776626; PMCID: PMC8579411.

30: Anderez DO, Kanjo E, Pogrebna G, Kaiwartya O, Johnson SD, Hunt JA. A

COVID-19-Based Modified Epidemiological Model and Technological Approaches to

Help Vulnerable Individuals Emerge from the Lockdown in the UK. Sensors (Basel).

2020 Sep 2;20(17):4967. doi: 10.3390/s20174967. PMID: 32887338; PMCID:

PMC7506567.

31: Deeb OE, Jalloul M. The dynamics of COVID-19 spread: evidence from Lebanon.

Math Biosci Eng. 2020 Aug 19;17(5):5618-5632. doi: 10.3934/mbe.2020302. PMID:

33120569.

32: Ahmad N, Qahmash A. A Hybrid Approach Toward COVID-19 Pandemic Modeling in

Saudi Arabia Using the Modified Susceptible-Exposed-Infectious-Recovered Model

and Open Data Sources. Cureus. 2021 Dec 8;13(12):e20279. doi:

10.7759/cureus.20279. PMID: 35028196; PMCID: PMC8748003.

33: Shen S, Li W, Wei H, Zhao L, Ye R, Ma K, Xiao P, Jia N, Zhou J, Cui X, Gong

J, Cao W. A Chess and Card Room-Induced COVID-19 Outbreak and Its Agent-Based

Simulation in Yangzhou, China. Front Public Health. 2022 Jun 17;10:915716. doi:

10.3389/fpubh.2022.915716. PMID: 35784212; PMCID: PMC9247329.

34: Alrashed S, Min-Allah N, Saxena A, Ali I, Mehmood R. Impact of lockdowns on

the spread of COVID-19 in Saudi Arabia. Inform Med Unlocked. 2020;20:100420.

doi: 10.1016/j.imu.2020.100420. Epub 2020 Sep 2. PMID: 32905098; PMCID:

PMC7462775.

35: Faraz N, Khan Y, Goufo EFD, Anjum A, Anjum A. Dynamic analysis of the

mathematical model of COVID-19 with demographic effects. Z Naturforsch C J

Biosci. 2020 Nov 26;75(11-12):389-396. doi: 10.1515/znc-2020-0121. PMID:

32920544.

36: Pan W, Li T, Ali S. A fractional order epidemic model for the simulation of

outbreaks of Ebola. Adv Differ Equ. 2021;2021(1):161. doi:

10.1186/s13662-021-03272-5. Epub 2021 Mar 10. PMID: 33719356; PMCID: PMC7943714.

37: Tay BK, Roby CA, Wu JW, Tan DY. Dynamical Analysis of Universal Masking on

the Pandemic. Int J Environ Res Public Health. 2021 Aug 27;18(17):9027. doi:

10.3390/ijerph18179027. PMID: 34501616; PMCID: PMC8430774.

38: Muller K, Muller PA. Mathematical modelling of the spread of COVID-19 on a

university campus. Infect Dis Model. 2021;6:1025-1045. doi:

10.1016/j.idm.2021.08.004. Epub 2021 Aug 14. PMID: 34414342; PMCID: PMC8364150.

39: Smirnova A, deCamp L, Chowell G. Forecasting Epidemics Through Nonparametric

Estimation of Time-Dependent Transmission Rates Using the SEIR Model. Bull Math

Biol. 2019 Nov;81(11):4343-4365. doi: 10.1007/s11538-017-0284-3. Epub 2017 May

2. PMID: 28466232.

40: Vyhmeister E, Provan G, Doyle B, Bourke B. Multi-cluster and environmental

dependant vector born disease models. Heliyon. 2020 Sep 1;6(9):e04090. doi:

10.1016/j.heliyon.2020.e04090. PMID: 32939408; PMCID: PMC7479329.

41: Stanovov V, Grabljevec S, Akhmedova S, Semenkin E, Stojanović R, Rozman Č,

Škraba A. Identification of COVID-19 spread mechanisms based on first-wave data,

simulation models, and evolutionary algorithms. PLoS One. 2022 Dec

28;17(12):e0279427. doi: 10.1371/journal.pone.0279427. PMID: 36576938; PMCID:

PMC9797101.

42: Berec L, Diviák T, Kuběna A, Levínský R, Neruda R, Suchopárová G, Šlerka J,

Šmíd M, Trnka J, Tuček V, Vidnerová P, Zajíček M. On the contact tracing for

COVID-19: A simulation study. Epidemics. 2023 Mar 16;43:100677. doi:

10.1016/j.epidem.2023.100677. Epub ahead of print. PMID: 36989916; PMCID:

PMC10019035.

43: Lyra W, do Nascimento JD Jr, Belkhiria J, de Almeida L, Chrispim PPM, de

Andrade I. COVID-19 pandemics modeling with modified determinist SEIR, social

distancing, and age stratification. The effect of vertical confinement and

release in Brazil. PLoS One. 2020 Sep 2;15(9):e0237627. doi:

10.1371/journal.pone.0237627. PMID: 32877420; PMCID: PMC7467267.

44: Ru D, Wen H, Zhang Y. A Pre-Generation of Emergency Reference Plan Model of

Public Health Emergencies with Case-Based Reasoning. Risk Manag Healthc Policy.

2022 Dec 15;15:2371-2388. doi: 10.2147/RMHP.S385967. PMID: 36544507; PMCID:

PMC9762414.

45: Calabrese JM, Demers J. How optimal allocation of limited testing capacity

changes epidemic dynamics. J Theor Biol. 2022 Apr 7;538:111017. doi:

10.1016/j.jtbi.2022.111017. Epub 2022 Jan 24. PMID: 35085536; PMCID: PMC8785410.

46: Cazelles B, Nguyen-Van-Yen B, Champagne C, Comiskey C. Dynamics of the

COVID-19 epidemic in Ireland under mitigation. BMC Infect Dis. 2021 Aug

3;21(1):735. doi: 10.1186/s12879-021-06433-9. PMID: 34344318; PMCID: PMC8329614.

47: Getz WM, Salter R, Luisa Vissat L, Koopman JS, Simon CP. A runtime alterable

epidemic model with genetic drift, waning immunity and vaccinations. J R Soc

Interface. 2021 Nov;18(184):20210648. doi: 10.1098/rsif.2021.0648. Epub 2021 Nov

24. PMID: 34814729; PMCID: PMC8611333.

48: Niakan Kalhori SR, Ghazisaeedi M, Azizi R, Naserpour A. Studying the

influence of mass media and environmental factors on influenza virus

transmission in the US Midwest. Public Health. 2019 May;170:17-22. doi:

10.1016/j.puhe.2019.02.006. Epub 2019 Mar 20. PMID: 30901605.

49: Li BZ, Cao NW, Zhou HY, Chu XJ, Ye DQ. Strong policies control the spread of

COVID-19 in China. J Med Virol. 2020 Oct;92(10):1980-1987. doi:

10.1002/jmv.25934. Epub 2020 Jul 28. PMID: 32330295; PMCID: PMC7264602.

50: Halley JM, Vokou D, Pappas G, Sainis I. SARS-CoV-2 mutational cascades and

the risk of hyper-exponential growth. Microb Pathog. 2021 Dec;161(Pt A):105237.

doi: 10.1016/j.micpath.2021.105237. Epub 2021 Oct 12. PMID: 34653544; PMCID:

PMC8507571.

51: Palaniappan S, V R, David B, S PN. Prediction of Epidemic Disease Dynamics

on the Infection Risk Using Machine Learning Algorithms. SN Comput Sci.

2022;3(1):47. doi: 10.1007/s42979-021-00902-3. Epub 2021 Nov 5. PMID: 34755116;

PMCID: PMC8570232.

52: Lin L, Chen B, Zhao Y, Wang W, He D. Two waves of COIVD-19 in Brazilian

cities and vaccination impact. Math Biosci Eng. 2022 Mar 9;19(5):4657-4671. doi:

10.3934/mbe.2022216. PMID: 35430833.

53: Sainz-Pardo JL, Valero J. COVID-19 and other viruses: Holding back its

spreading by massive testing. Expert Syst Appl. 2021 Dec 30;186:115710. doi:

10.1016/j.eswa.2021.115710. Epub 2021 Aug 9. PMID: 34393387; PMCID: PMC8351357.

54: Ding Y, Gao L. An evaluation of COVID-19 in Italy: A data-driven modeling

analysis. Infect Dis Model. 2020 Jul 9;5:495-501. doi:

10.1016/j.idm.2020.06.007. PMID: 32766461; PMCID: PMC7347309.

55: Gao J, Zhou C, Liang H, Jiao R, Wheelock √ÖM, Jiao K, Ma J, Zhang C, Guo Y,

Luo S, Liang W, Xu L. Monkeypox outbreaks in the context of the COVID-19

pandemic: Network and clustering analyses of global risks and modified SEIR

prediction of epidemic trends. Front Public Health. 2023 Jan 24;11:1052946. doi:

10.3389/fpubh.2023.1052946. PMID: 36761122; PMCID: PMC9902715.

56: Zheng Y, Wang Y. How Seasonality and Control Measures Jointly Determine the

Multistage Waves of the COVID-19 Epidemic: A Modelling Study and Implications.

Int J Environ Res Public Health. 2022 May 25;19(11):6404. doi:

10.3390/ijerph19116404. PMID: 35681989; PMCID: PMC9180569.

57: Liu M, Ning J, Du Y, Cao J, Zhang D, Wang J, Chen M. Modelling the evolution

trajectory of COVID-19 in Wuhan, China: experience and suggestions. Public

Health. 2020 Jun;183:76-80. doi: 10.1016/j.puhe.2020.05.001. Epub 2020 May 12.

PMID: 32442842; PMCID: PMC7214341.

58: Chang YS, Mayer S, Davis ES, Figueroa E, Leo P, Finn PW, Perkins DL.

Transmission Dynamics of Large Coronavirus Disease Outbreak in Homeless Shelter,

Chicago, Illinois, USA, 2020. Emerg Infect Dis. 2022 Jan;28(1):76-84. doi:

10.3201/eid2801.210780. Epub 2021 Dec 2. PMID: 34856112; PMCID: PMC8714208.

59: Acheampong E, Okyere E, Iddi S, Bonney JHK, Asamoah JKK, Wattis JAD, Gomes

RL. Mathematical modelling of earlier stages of COVID-19 transmission dynamics

in Ghana. Results Phys. 2022 Mar;34:105193. doi: 10.1016/j.rinp.2022.105193.

Epub 2022 Jan 14. PMID: 35070648; PMCID: PMC8759145.

60: Arfan M, Alrabaiah H, Rahman MU, Sun YL, Hashim AS, Pansera BA, Ahmadian A,

Salahshour S. Investigation of fractal-fractional order model of COVID-19 in

Pakistan under Atangana-Baleanu Caputo (ABC) derivative. Results Phys. 2021

May;24:104046. doi: 10.1016/j.rinp.2021.104046. Epub 2021 Mar 22. PMID:

33868907; PMCID: PMC8044634.

61: Ala'raj M, Majdalawieh M, Nizamuddin N. Modeling and forecasting of COVID-19

using a hybrid dynamic model based on SEIRD with ARIMA corrections. Infect Dis

Model. 2021;6:98-111. doi: 10.1016/j.idm.2020.11.007. Epub 2020 Dec 3. PMID:

33294749; PMCID: PMC7713640.

62: Alsinglawi B, Mubin O, Alnajjar F, Kheirallah K, Elkhodr M, Al Zobbi M,

Novoa M, Arsalan M, Poly TN, Gochoo M, Khan G, Dev K. A simulated measurement

for COVID-19 pandemic using the effective reproductive number on an empirical

portion of population: epidemiological models. Neural Comput Appl. 2021 Oct

9:1-9. doi: 10.1007/s00521-021-06579-2. Epub ahead of print. PMID: 34658535;

PMCID: PMC8502096.

63: Liu X, Huang J, Li C, Zhao Y, Wang D, Huang Z, Yang K. The role of

seasonality in the spread of COVID-19 pandemic. Environ Res. 2021

Apr;195:110874. doi: 10.1016/j.envres.2021.110874. Epub 2021 Feb 19. PMID:

33610582; PMCID: PMC7892320.

64: Kumar S, Xu C, Ghildayal N, Chandra C, Yang M. Social media effectiveness as

a humanitarian response to mitigate influenza epidemic and COVID-19 pandemic.

Ann Oper Res. 2022;319(1):823-851. doi: 10.1007/s10479-021-03955-y. Epub 2021

Jan 29. PMID: 33531729; PMCID: PMC7843901.

65: Cheng T, Liu J, Liu Y, Zhang X, Gao X. Measures to prevent nosocomial

transmissions of COVID-19 based on interpersonal contact data. Prim Health Care

Res Dev. 2022 Jan 28;23:e4. doi: 10.1017/S1463423621000852. PMID: 35086594;

PMCID: PMC8822327.

66: Shi P, Dong Y, Yan H, Zhao C, Li X, Liu W, He M, Tang S, Xi S. Impact of

temperature on the dynamics of the COVID-19 outbreak in China. Sci Total

Environ. 2020 Aug 1;728:138890. doi: 10.1016/j.scitotenv.2020.138890. Epub 2020

Apr 23. PMID: 32339844; PMCID: PMC7177086.

67: Kwok WC, Wong CK, Ma TF, Ho KW, Fan LW, Chan KF, Chan SS, Tam TC, Ho PL.

Modelling the impact of travel restrictions on COVID-19 cases in Hong Kong in

early 2020. BMC Public Health. 2021 Oct 18;21(1):1878. doi:

10.1186/s12889-021-11889-0. Erratum in: BMC Public Health. 2021 Nov

17;21(1):2115. PMID: 34663279; PMCID: PMC8522545.

68: Canga A, Bidegain G. Modelling the Effect of the Interaction between

Vaccination and Nonpharmaceutical Measures on COVID-19 Incidence. Glob Health

Epidemiol Genom. 2022 Mar 31;2022:9244953. doi: 10.1155/2022/9244953. PMID:

35392137; PMCID: PMC8968356.

69: Pei Y, Li J, Xu S, Xu Y. Adaptive Multi-Factor Quantitative Analysis and

Prediction Models: Vaccination, Virus Mutation and Social Isolation on COVID-19.

Front Med (Lausanne). 2022 Mar 16;9:828691. doi: 10.3389/fmed.2022.828691. PMID:

35372421; PMCID: PMC8965859.

70: Kinyili M, Munyakazi JB, Mukhtar AY. Mathematical modeling and impact

analysis of the use of COVID Alert SA app. AIMS Public Health. 2021 Nov

29;9(1):106-128. doi: 10.3934/publichealth.2022009. PMID: 35071672; PMCID:

PMC8755967.

71: Suwantika AA, Dhamanti I, Suharto Y, Purba FD, Abdulah R. The cost-

effectiveness of social distancing measures for mitigating the COVID-19 pandemic

in a highly-populated country: A case study in Indonesia. Travel Med Infect Dis.

2022 Jan-Feb;45:102245. doi: 10.1016/j.tmaid.2021.102245. Epub 2021 Dec 23.

PMID: 34954344; PMCID: PMC8695594.

72: Herng LC, Singh S, Sundram BM, Zamri ASSM, Vei TC, Aris T, Ibrahim H,

Abdullah NH, Dass SC, Gill BS. The effects of super spreading events and

movement control measures on the COVID-19 pandemic in Malaysia. Sci Rep. 2022

Feb 9;12(1):2197. doi: 10.1038/s41598-022-06341-1. PMID: 35140319; PMCID:

PMC8828893.

73: Shlomai A, Leshno A, Sklan EH, Leshno M. Modeling Social Distancing

Strategies to Prevent SARS-CoV-2 Spread in Israel: A Cost-Effectiveness

Analysis. Value Health. 2021 May;24(5):607-614. doi: 10.1016/j.jval.2020.09.013.

Epub 2020 Dec 9. PMID: 33933228; PMCID: PMC7833124.

74: Nakhaeizadeh M, Eybpoosh S, Jahani Y, Ahmadi Gohari M, Haghdoost AA, White

L, Sharifi H. Impact of Non-pharmaceutical Interventions on the Control of

COVID-19 in Iran: A Mathematical Modeling Study. Int J Health Policy Manag. 2021

Jun 9;11(8):1472–81. doi: 10.34172/ijhpm.2021.48. Epub ahead of print. PMID:

34273920; PMCID: PMC9808365.

75: Yousif A, Ali A. The impact of intervention strategies and prevention

measurements for controlling COVID-19 outbreak in Saudi Arabia. Math Biosci Eng.

2020 Nov 13;17(6):8123-8137. doi: 10.3934/mbe.2020412. PMID: 33378936.

76: Motisi N, Papaïx J, Poggi S. The Dark Side of Shade: How Microclimates Drive

the Epidemiological Mechanisms of Coffee Berry Disease. Phytopathology. 2022

Jun;112(6):1235-1243. doi: 10.1094/PHYTO-06-21-0247-R. Epub 2022 May 2. PMID:

35505280.

77: Nakhaeizadeh M, Chegeni M, Adhami M, Sharifi H, Gohari MA, Iranpour A,

Azizian M, Mashinchi M, Baneshi MR, Karamouzian M, Haghdoost AA, Jahani Y.

Estimating the Number of COVID-19 Cases and Impact of New COVID-19 Variants and

Vaccination on the Population in Kerman, Iran: A Mathematical Modeling Study.

Comput Math Methods Med. 2022 Apr 26;2022:6624471. doi: 10.1155/2022/6624471.

PMID: 35495892; PMCID: PMC9039779.

78: Thron C, Mbazumutima V, Tamayo LV, Todjihounde L. Cost effective

reproduction number based strategies for reducing deaths from COVID-19. J Math

Ind. 2021;11(1):11. doi: 10.1186/s13362-021-00107-6. Epub 2021 Jun 28. PMID:

34221823; PMCID: PMC8237561.

79: Zelner JL, Lopman BA, Hall AJ, Ballesteros S, Grenfell BT. Linking time-

varying symptomatology and intensity of infectiousness to patterns of norovirus

transmission. PLoS One. 2013 Jul 24;8(7):e68413. doi:

10.1371/journal.pone.0068413. PMID: 23894302; PMCID: PMC3722229.

80: Kamrujjaman M, Mahmud MS, Ahmed S, Qayum MO, Alam MM, Hassan MN, Islam MR,

Nipa KF, Bulut U. SARS-CoV-2 and Rohingya Refugee Camp, Bangladesh: Uncertainty

and How the Government Took Over the Situation. Biology (Basel). 2021 Feb

5;10(2):124. doi: 10.3390/biology10020124. PMID: 33562509; PMCID: PMC7914953.

81: López L, Rodó X. The end of social confinement and COVID-19 re-emergence

risk. Nat Hum Behav. 2020 Jul;4(7):746-755. doi: 10.1038/s41562-020-0908-8. Epub

2020 Jun 22. PMID: 32572175.

82: Quiner C, Jones K, Bobashev G. Impacts of timing, length, and intensity of

behavioral interventions to COVID-19 dynamics: North Carolina county-level

examples. Infect Dis Model. 2022 Sep;7(3):535-544. doi:

10.1016/j.idm.2022.08.002. Epub 2022 Aug 13. PMID: 35992738; PMCID: PMC9374497.

83: Aljuboury AS, Abedi F, Shukur HM, Hashim ZS, Ibraheem IK, Alkhayyat A.

Mathematical Modeling and Control of COVID-19 Using Super Twisting Sliding Mode

and Nonlinear Techniques. Comput Intell Neurosci. 2022 Jun 30;2022:8539278. doi:

10.1155/2022/8539278. PMID: 35785071; PMCID: PMC9244765.

84: Yu Z, Zhu X, Liu X, Wei T, Yuan HY, Xu Y, Zhu R, He H, Wang H, Wong MS, Jia

P, Guo S, Shi W, Chen W. Reopening International Borders without Quarantine:

Contact Tracing Integrated Policy against COVID-19. Int J Environ Res Public

Health. 2021 Jul 14;18(14):7494. doi: 10.3390/ijerph18147494. PMID: 34299945;

PMCID: PMC8303901.

85: Pei Y, Guo Y, Wu T, Liang H. Quantifying the dynamic transmission of

COVID-19 asymptomatic and symptomatic infections: Evidence from four Chinese

regions. Front Public Health. 2022 Sep 29;10:925492. doi:

10.3389/fpubh.2022.925492. PMID: 36249263; PMCID: PMC9557086.

86: Singh A, Deolia P. COVID-19 outbreak: a predictive mathematical study

incorporating shedding effect. J Appl Math Comput. 2023;69(1):1239-1268. doi:

10.1007/s12190-022-01792-1. Epub 2022 Sep 19. PMID: 36158635; PMCID: PMC9484852.

87: Yan Q, Tang Y, Yan D, Wang J, Yang L, Yang X, Tang S. Impact of media

reports on the early spread of COVID-19 epidemic. J Theor Biol. 2020 Oct

7;502:110385. doi: 10.1016/j.jtbi.2020.110385. Epub 2020 Jun 25. PMID: 32593679;

PMCID: PMC7316072.

88: Narouei-Khandan HA, Shakya SK, Garrett KA, Goss EM, Dufault NS, Andrade-

Piedra JL, Asseng S, Wallach D, Bruggen AHCV. BLIGHTSIM: A New Potato Late

Blight Model Simulating the Response of <i>Phytophthora infestans</i> to Diurnal

Temperature and Humidity Fluctuations in Relation to Climate Change. Pathogens.

2020 Aug 15;9(8):659. doi: 10.3390/pathogens9080659. PMID: 32824250; PMCID:

PMC7459445.

89: Chatterjee K, Shankar S, Chatterjee K, Yadav AK. Coronavirus disease 2019 in

India: Post-lockdown scenarios and provisioning for health care. Med J Armed

Forces India. 2020 Oct;76(4):387-394. doi: 10.1016/j.mjafi.2020.06.004. Epub

2020 Jun 18. PMID: 32836711; PMCID: PMC7301789.

90: Asgary A, Cojocaru MG, Najafabadi MM, Wu J. Simulating preventative testing

of SARS-CoV-2 in schools: policy implications. BMC Public Health. 2021 Jan

12;21(1):125. doi: 10.1186/s12889-020-10153-1. PMID: 33430832; PMCID:

PMC7801157.

91: Zhang S, Ponce J, Zhang Z, Lin G, Karniadakis G. An integrated framework for

building trustworthy data-driven epidemiological models: Application to the

COVID-19 outbreak in New York City. PLoS Comput Biol. 2021 Sep 8;17(9):e1009334.

doi: 10.1371/journal.pcbi.1009334. PMID: 34495965; PMCID: PMC8452065.

92: Maji A, Choudhari T, Sushma MB. Implication of repatriating migrant workers

on COVID-19 spread and transportation requirements. Transp Res Interdiscip

Perspect. 2020 Sep;7:100187. doi: 10.1016/j.trip.2020.100187. Epub 2020 Aug 3.

PMID: 34173463; PMCID: PMC7396945.

93: Obeng-Kusi M, Habila MA, Roe DJ, Erstad B, Abraham I. Economic evaluation

using dynamic transition modeling of ebola virus vaccination in lower-and-

middle-income countries. J Med Econ. 2021 Nov;24(sup1):1-13. doi:

10.1080/13696998.2021.2002092. Update in: J Med Econ. 2022 Jan-

Dec;25(1):894-902. PMID: 34866541.

94: Ghosh SK, Ghosh S. A mathematical model for COVID-19 considering waning

immunity, vaccination and control measures. Sci Rep. 2023 Mar 3;13(1):3610. doi:

10.1038/s41598-023-30800-y. PMID: 36869104; PMCID: PMC9983535.

95: Ng KY, Gui MM. COVID-19: Development of a robust mathematical model and

simulation package with consideration for ageing population and time delay for

control action and resusceptibility. Physica D. 2020 Oct;411:132599. doi:

10.1016/j.physd.2020.132599. Epub 2020 Jun 9. PMID: 32536738; PMCID: PMC7282799.

96: Abdulrahman I. SimCOVID: Open-Source Simulation Programs for the COVID-19

Outbreak. SN Comput Sci. 2023;4(1):20. doi: 10.1007/s42979-022-01441-1. Epub

2022 Oct 19. PMID: 36274814; PMCID: PMC9580422.

97: Zhang WB, Ge Y, Liu M, Atkinson PM, Wang J, Zhang X, Tian Z. Risk assessment

of the step-by-step return-to-work policy in Beijing following the COVID-19

epidemic peak. Stoch Environ Res Risk Assess. 2021;35(2):481-498. doi:

10.1007/s00477-020-01929-3. Epub 2020 Nov 13. PMID: 33223954; PMCID: PMC7664171.

98: Getz WM, Salter R, Luisa Vissat L, Horvitz N. A versatile web app for

identifying the drivers of COVID-19 epidemics. J Transl Med. 2021 Mar

16;19(1):109. doi: 10.1186/s12967-021-02736-2. PMID: 33726787; PMCID:

PMC7962635.

99: He J, Chen G, Jiang Y, Jin R, Shortridge A, Agusti S, He M, Wu J, Duarte CM,

Christakos G. Comparative infection modeling and control of COVID-19

transmission patterns in China, South Korea, Italy and Iran. Sci Total Environ.

2020 Dec 10;747:141447. doi: 10.1016/j.scitotenv.2020.141447. Epub 2020 Aug 3.

PMID: 32771775; PMCID: PMC7397934.

100: Zhao H, Feng Z. Staggered release policies for COVID-19 control: Costs and

benefits of relaxing restrictions by age and risk. Math Biosci. 2020

Aug;326:108405. doi: 10.1016/j.mbs.2020.108405. Epub 2020 Jun 18. PMID:

32565231; PMCID: PMC7301890.

101: Wickramaarachchi WPTM, Perera SSN, Jayasinghe S. COVID-19 Epidemic in Sri

Lanka: A Mathematical and Computational Modelling Approach to Control. Comput

Math Methods Med. 2020 Oct 16;2020:4045064. doi: 10.1155/2020/4045064. PMID:

33101453; PMCID: PMC7573659.

102: Schuette MC. A qualitative analysis of a model for the transmission of

varicella-zoster virus. Math Biosci. 2003 Apr;182(2):113-26. doi:

10.1016/s0025-5564(02)00219-5. PMID: 12591619.

103: Kheirallah KA, Alsinglawi B, Alzoubi A, Saidan MN, Mubin O, Alorjani MS,

Mzayek F. The Effect of Strict State Measures on the Epidemiologic Curve of

COVID-19 Infection in the Context of a Developing Country: A Simulation from

Jordan. Int J Environ Res Public Health. 2020 Sep 8;17(18):6530. doi:

10.3390/ijerph17186530. PMID: 32911738; PMCID: PMC7558493.

104: Liu PY, He S, Rong LB, Tang SY. The effect of control measures on COVID-19

transmission in Italy: Comparison with Guangdong province in China. Infect Dis

Poverty. 2020 Sep 16;9(1):130. doi: 10.1186/s40249-020-00730-2. PMID: 32938502;

PMCID: PMC7492796.

105: Topîrceanu A. Immunization using a heterogeneous geo-spatial population

model: A qualitative perspective on COVID-19 vaccination strategies. Procedia

Comput Sci. 2021;192:2095-2104. doi: 10.1016/j.procs.2021.08.217. Epub 2021 Oct

1. PMID: 34630745; PMCID: PMC8486231.

106: Khadadah F, Al-Shammari AA, Alhashemi A, Alhuwail D, Al-Saif B, Alzaid SN,

Alahmad B, Bogoch II. The effects of non-pharmaceutical interventions on SARS-

CoV-2 transmission in different socioeconomic populations in Kuwait: a modeling

study. BMC Public Health. 2021 May 26;21(1):990. doi:

10.1186/s12889-021-10984-6. PMID: 34039289; PMCID: PMC8152192.

107: Chen T, Huang S, Li G, Zhang Y, Li Y, Zhu J, Shi X, Li X, Xie G, Zhang L.

An integrated framework for modelling quantitative effects of entry restrictions

and travel quarantine on importation risk of COVID-19. J Biomed Inform. 2021

Jun;118:103800. doi: 10.1016/j.jbi.2021.103800. Epub 2021 May 7. PMID: 33965636;

PMCID: PMC8102072.

108: Glasser JW, Feng Z, Vo M, Jones JN, Clarke KEN. Analysis of serological

surveys of antibodies to SARS-CoV-2 in the United States to estimate parameters

needed for transmission modeling and to evaluate and improve the accuracy of

predictions. J Theor Biol. 2023 Jan 7;556:111296. doi:

10.1016/j.jtbi.2022.111296. Epub 2022 Oct 5. PMID: 36208669; PMCID: PMC9532270.

109: Costantino V, Kunasekaran M, MacIntyre CR. Modelling of optimal vaccination

strategies in response to a bioterrorism associated smallpox outbreak. Hum

Vaccin Immunother. 2021 Mar 4;17(3):738-746. doi: 10.1080/21645515.2020.1800324.

Epub 2020 Dec 2. PMID: 33734944; PMCID: PMC7993194.

110: Yu X. Modeling return of the epidemic: Impact of population structure,

asymptomatic infection, case importation and personal contacts. Travel Med

Infect Dis. 2020 Sep-Oct;37:101858. doi: 10.1016/j.tmaid.2020.101858. Epub 2020

Aug 27. PMID: 32860959; PMCID: PMC7449940.

111: Zeng Z, Qu W, Liu R, Guan W, Liang J, Lin Z, Lau EHY, Hon C, Yang Z, He J.

Real-time assessment of COVID-19 epidemic in Guangdong Province, China using

mathematical models. J Thorac Dis. 2023 Mar 31;15(3):1517-1522. doi:

10.21037/jtd-23-47. Epub 2023 Mar 20. PMID: 37065584; PMCID: PMC10089860.

112: Yan Q, Tang S, Gabriele S, Wu J. Media coverage and hospital notifications:

Correlation analysis and optimal media impact duration to manage a pandemic. J

Theor Biol. 2016 Feb 7;390:1-13. doi: 10.1016/j.jtbi.2015.11.002. Epub 2015 Nov

12. PMID: 26582723.

113: Nielsen JP, Larsen TS, Halasa T, Christiansen LE. Estimation of the

transmission dynamics of African swine fever virus within a swine house.

Epidemiol Infect. 2017 Oct;145(13):2787-2796. doi: 10.1017/S0950268817001613.

Epub 2017 Aug 3. PMID: 28768556; PMCID: PMC9148809.

114: Hu B, Jiang G, Yao X, Chen W, Yue T, Zhao Q, Wen Z. Allocation of emergency

medical resources for epidemic diseases considering the heterogeneity of

epidemic areas. Front Public Health. 2023 Feb 24;11:992197. doi:

10.3389/fpubh.2023.992197. PMID: 36908482; PMCID: PMC9998515.

115: Ball FG, Knock ES, O'Neill PD. Stochastic epidemic models featuring contact

tracing with delays. Math Biosci. 2015 Aug;266:23-35. doi:

10.1016/j.mbs.2015.05.007. Epub 2015 May 30. PMID: 26037511.

116: Verma VR, Saini A, Gandhi S, Dash U, Koya SF. Capacity-need gap in hospital

resources for varying mitigation and containment strategies in India in the face

of COVID-19 pandemic. Infect Dis Model. 2020;5:608-621. doi:

10.1016/j.idm.2020.08.011. Epub 2020 Aug 28. PMID: 32875175; PMCID: PMC7452840.

117: Zhao Z, Li X, Liu F, Zhu G, Ma C, Wang L. Prediction of the COVID-19 spread

in African countries and implications for prevention and control: A case study

in South Africa, Egypt, Algeria, Nigeria, Senegal and Kenya. Sci Total Environ.

2020 Aug 10;729:138959. doi: 10.1016/j.scitotenv.2020.138959. Epub 2020 Apr 25.

PMID: 32375067; PMCID: PMC7182531.

118: Ejigu BA, Asfaw MD, Cavalerie L, Abebaw T, Nanyingi M, Baylis M. Assessing

the impact of non-pharmaceutical interventions (NPI) on the dynamics of

COVID-19: A mathematical modelling study of the case of Ethiopia. PLoS One. 2021

Nov 16;16(11):e0259874. doi: 10.1371/journal.pone.0259874. PMID: 34784379;

PMCID: PMC8594814.

119: Wang XL, Lin X, Yang P, Wu ZY, Li G, McGoogan JM, Jiao ZT, He XJ, Li SQ,

Shi HH, Wang JY, Lai SJ, Huang C, Wang QY. Coronavirus disease 2019 outbreak in

Beijing's Xinfadi Market, China: a modeling study to inform future resurgence

response. Infect Dis Poverty. 2021 May 7;10(1):62. doi:

10.1186/s40249-021-00843-2. PMID: 33962683; PMCID: PMC8103671.

120: Kinyili M, Munyakazi JB, Mukhtar AYA. Modeling the impact of combined use

of COVID Alert SA app and vaccination to curb COVID-19 infections in South

Africa. PLoS One. 2023 Feb 3;18(2):e0264863. doi: 10.1371/journal.pone.0264863.

PMID: 36735664; PMCID: PMC9897588.

121: Candel FJ, Viayna E, Callejo D, Ramos R, San-Roman-Montero J, Barreiro P,

Carretero MDM, Kolipiński A, Canora J, Zapatero A, Runken MC. Social

Restrictions versus Testing Campaigns in the COVID-19 Crisis: A Predictive Model

Based on the Spanish Case. Viruses. 2021 May 15;13(5):917. doi:

10.3390/v13050917. PMID: 34063465; PMCID: PMC8157049.

122: Zhang JW, Han PE, Yang L. [Spatial accessibility of fever clinics for

multi-tiered prevention and control on COVID-19 in Beijing]. Beijing Da Xue Xue

Bao Yi Xue Ban. 2021 Jun 18;53(3):543-548. Chinese. doi:

10.19723/j.issn.1671-167X.2021.03.017. PMID: 34145858; PMCID: PMC8220033.

123: Park AW, Wood JL, Daly JM, Newton JR, Glass K, Henley W, Mumford JA,

Grenfell BT. The effects of strain heterology on the epidemiology of equine

influenza in a vaccinated population. Proc Biol Sci. 2004 Aug

7;271(1548):1547-55. doi: 10.1098/rspb.2004.2766. PMID: 15306299; PMCID:

PMC1691760.

124: Mugdha SBS, Uddin M, Islam MT. Extended Epidemiological Models for Weak

Economic Region: Case Studies of the Spreading of COVID-19 in the South Asian

Subcontinental Countries. Biomed Res Int. 2021 Oct 19;2021:7787624. doi:

10.1155/2021/7787624. PMID: 34676263; PMCID: PMC8526245.

125: Farber DH, De Leenheer P, Mundt CC. Dispersal Kernels may be Scalable:

Implications from a Plant Pathogen. J Biogeogr. 2019 Sep;46(9):2042-2055. doi:

10.1111/jbi.13642. Epub 2019 Jul 2. PMID: 33041433; PMCID: PMC7546428.

126: Deguen S, Flahault A. Impact on immunization of seasonal cycle of

chickenpox. Eur J Epidemiol. 2000;16(12):1177-81. doi: 10.1023/a:1010935317612.

PMID: 11484809.

127: Cai QC, Jiang QW, Xu QF, Cheng X, Guo Q, Sun QW, Zhao GM. [To develop a

model on severe acute respiratory syndrome epidemics to quantitatively evaluate

the effectiveness of intervention measures]. Zhonghua Liu Xing Bing Xue Za Zhi.

2005 Mar;26(3):153-8. Chinese. PMID: 15941495.

128: Awad K, Moore L, Huang J, Gomez L, Brotto L, Varanasi VG, Cardozo C,

Weisleder N, Pan Z, Zhou J, Bonewald L, Brotto M. Advanced Methodology for Rapid

Isolation of Single Myofibers from Flexor Digitorum Brevis Muscle. Tissue Eng

Part C Methods. 2023 Apr 25. doi: 10.1089/ten.TEC.2023.0012. Epub ahead of

print. PMID: 37097213.

129: Takeuchi S, Kuroda Y. [Predicting spread of new pandemic swine-origin

influenza A (H1N1) in local mid-size city: evaluation of hospital bed shortage

and effectiveness of vaccination]. Nihon Eiseigaku Zasshi. 2010 Jan;65(1):48-52.

Japanese. doi: 10.1265/jjh.65.48. PMID: 20134108.

A few grammar errors and contextual presentations need revision.

Round 2

Reviewer 2 Report

References and explanations are updated as suggested.